# Impact of COVID-19 Disruptions on Global BCG Coverage and Paediatric TB Mortality: A Modelling Study

**DOI:** 10.3390/vaccines9111228

**Published:** 2021-10-22

**Authors:** Nabila Shaikh, Puck T. Pelzer, Sanne M. Thysen, Partho Roy, Rebecca C. Harris, Richard G. White

**Affiliations:** 1TB Modelling Group, TB Centre, Faculty of Epidemiology and Population Health, London School of Hygiene and Tropical Medicine, London WC1E 7HT, UK; Rebecca.harris@lshtm.ac.uk (R.C.H.); Richard.white@lshtm.ac.uk (R.G.W.); 2Technical Division, KNCV Tuberculosis, Maanweg 174, 2516 AB The Hague, The Netherlands; puck.pelzer@kncvtbc.org; 3Center for Clinical Research and Prevention, Bispebjerg and Frederiksberg Hospital, 2004 Frederiksberg, Denmark; samt@dadlnet.dk; 4Bandim Health Project, Apartado 861, Bissau 1004, Guinea-Bissau; 5Immunisation and Countermeasures, National Infection Service, Public Health England, London NW9 5EQ, UK; Partho.Roy@phe.gov.uk; 6Sanofi Pasteur, South Beach Tower 18-11, Singapore 189767, Singapore

**Keywords:** BCG, immunization, mathematical model, paediatric mortality, COVID-19, tuberculosis

## Abstract

The impact of COVID-19 disruptions on global Bacillus Calmette-Guérin (BCG) coverage and paediatric tuberculosis (TB) mortality is still unknown. To fill this evidence-gap and guide mitigation measures, we estimated the impact of COVID-19 disruptions on global BCG coverage and paediatric TB mortality. First, we used data from multiple sources to estimate COVID-19-disrupted BCG vaccination coverage. Second, using a static mathematical model, we estimated the number of additional paediatric TB deaths in the first 15 years of life due to delayed/missed vaccinations in 14 scenarios—varying in duration of disruption, and magnitude and timing of catch-up. We estimated a 25% reduction in global BCG coverage within the disruption period. The best-case scenario (3-month disruption, 100% catch-up within 3 months) resulted in an additional 886 (0.5%) paediatric TB deaths, and the worst-case scenario (6-month disruption with no catch-up) resulted in an additional 33,074 (17%) deaths. The magnitude of catch-up was found to be the most influential variable in minimising excess paediatric TB mortality. Our results show that ensuring catch-up vaccination of missed children is a critical priority, and delivery of BCG alongside other routine vaccines may be a feasible way to achieve catch-up. Urgent action is required to support countries with recovering vaccination coverages to minimise paediatric deaths.

## 1. Introduction

On 22 May 2020, Gavi (the Vaccine Alliance), the United Nations Children’s Fund (UNICEF) and the World Health Organisation (WHO) warned of approximately 80 million children at risk of missing their childhood vaccination [1,2]. Transport delays and logistical problems threaten vaccine shipments, making low- and middle-income countries (LMIC) especially vulnerable to stockouts and insufficient resources for maintaining vaccine programs [3]. The oldest childhood vaccine included in most LMIC immunisation programs is the Bacillus Calmette-Guérin (BCG) vaccine. In 2021, the BCG vaccine celebrates its 100-year anniversary; it remains the only approved vaccine against tuberculosis (TB) [4]. The efficacy of the BCG vaccine against TB has varied in different studies and populations, nevertheless, the preventive effect of BCG against miliary TB and TB meningitis is well documented, particularly in children [5]. Global BCG vaccination policy varies substantially by country, and currently 152 LMIC have a policy of universal neonatal BCG vaccination. The WHO recommendation is to administer BCG at birth, however, only 56% of eligible neonates receive BCG within the first month of life and 89% by 3 years of age [6]. A previous pre-COVID-19 modelling study estimated that reducing delays or increasing coverage of BCG at birth would substantially reduce global paediatric tuberculosis mortality [7].

During prior epidemics such as Ebola, BCG has been particularly impacted due to stockouts in affected regions [8]. As a result of the global BCG stockout in 2013–2015, studies estimated an additional 7433 paediatric TB deaths occurred in the affected birth cohort [9]. Multiple studies have begun to quantify the reduction in routine paediatric immunisations in various regions due to COVID-19 [10,11]. A systematic review of the impact of COVID-19-related disruption to immunisations found a significant drop in vaccine coverage in 2020, particularly noting reductions in measles, mumps, and rubella (MMR), poliomyelitis, and influenza vaccinations [2]. This review only included one specific study on BCG disruption, noting a 41% vaccine coverage decline in Pakistan [12]. A further multi-country study by Maresha et al. found there was substantial variation in impact of COVID-19 on routine immunization programs; some African countries had maintained BCG coverage while others showed lower rates between January and June 2020 [13]. 

Missed or delayed BCG vaccination leaves children vulnerable to TB disease. Previous studies have estimated that even briefly delayed BCG vaccinations may increase the burden of paediatric TB mortality [7,9] and country-specific modelling has found that utilizing health-facility contacts to provide BCG may reduce both TB and all-cause mortality [14]. 

Currently, there are limited data on the global reduction of BCG coverage caused by COVID-19-related disruptions, and the potential impact of such disruptions on paediatric TB mortality is still unknown. To fill this evidence gap, we conducted a two-stage analysis (1) to estimate the impact of COVID-19 disruptions on global BCG coverage, and (2) repurposed an existing model to estimate the additional paediatric TB mortality in LMICs with a universal BCG policy. The magnitude of these disruptions needs to be urgently quantified to prompt swift and effective remedial action by the global health community. While we focus on the impact of the COVID-19 pandemic on BCG disruptions, the method presented in this study may also be relevant for other childhood vaccines disrupted during this pandemic period.

## 2. Materials and Methods

### 2.1. BCG Disruption Data

We collated available data from peer-reviewed papers, newspaper articles, published country level reports, and conversations with experts to estimate the magnitude of global reduction in BCG coverage. Specifically, we searched for information on whether BCG was interrupted in-country due to COVID-19, and if so, the period of disruption and the percentage reduction compared to 2019 values. Where data indicated some disruption to BCG, but specific data were not available, we used estimates of disruption to other routinely delivered paediatric vaccines as a proxy.

### 2.2. BCG Disruption Magnitude 

The global estimate of relative BCG coverage reduction (R_2020_) within the disruption period in 2020 was estimated as follows:R_2020_ = ∑ (C ^i^_2020_ × W^i^)(1)
W^i^ = *N* ^i^_2019_ /G_2019_(2)
where the country level estimate of relative reduction in BCG coverage C ^i^_2020_ (percentage of BCG given in the disruption period in 2020 compared to the same period in 2019) in each country (i) was multiplied by a weighting W^i^. The weighting was calculated as the number of BCG doses (*N* ^i^_2019_) each country delivered in 2019 divided by the global number of doses of BCG provided in 2019 (G_2019_). We assumed that reduction estimates from countries where data were available, and our estimate of R_2020_, were representative and generalisable to all 152 LMIC with a policy for universal neonatal vaccination. R_2020_ was the final estimate of relative reduction in BCG coverage applied in the model. 

### 2.3. BCG Disruption Duration 

As there were extremely limited data on the duration of disruption, and estimates varied substantially by setting, we developed multiple scenarios with either a 3-month or 6-month duration of disruption to cover the range indicated by our review. Given that there were limited data available on country plans and priorities for immunisation catch-ups, we developed three assumptions for the timing of the catch-up strategies; (i) catch-up would occur linearly within 3 months following the end of the disruption period, (ii) catch-up would occur linearly within 6 months following the end of the disruption period (iii) catch-up was timed with the MMR vaccine (approximately 19 months) as a proxy for co-delivering BCG with other routine paediatric immunisations. The global MMR timeliness data were estimated by Clark et al., where 60% of eligible children receive the vaccine within 1 year and 90% by 2 years of age [6]. As global MMR coverage is higher than for BCG, to be conservative we capped the coverage in the MMR-timed scenarios to match the maximum coverage of BCG.

### 2.4. Populations Affected by Disruption

Pulse survey data showed that vaccinations at birth were least affected by COVID-19-related disruptions, and that outreach programmes were more affected than healthcare facilities [10].

Therefore, we assumed that vaccinations usually administered at birth in healthcare facilities were not disrupted, but vaccinations usually given at birth outside healthcare facilities, and all vaccinations usually given after birth could be disrupted. Using this, we estimated the proportion of the BCG-eligible cohort affected by COVID-19 disruption was 69% from UNICEF maternal and neonatal health data and Clark et al vaccination timing data [6,15].

Applying this, the number of vaccinated children each week in 2020 (V_t 2020_) was estimated as:V_t 2020_ = V_t 2019_ × P_2020_ × B_2020_ × (1 − R_2020_).(3)
where V_t 2019_ was the number of neonates who would have received the vaccine each week t in 2019 (i.e. no disruption), P_2020_ was the proportion of the BCG eligible cohort affected by COVID disruption (69%), B_2020_ was the percentage of neonates born during the disruption period (i.e. 25% of the birth cohort in the 3-month disruption scenarios and 50% of the birth cohort born in the 6-month disruption scenarios), and R_2020_ was relative reduction in BCG coverage level during disruption (25%).

### 2.5. Model Disruption Scenarios 

A total of 14 hypothetical scenarios were generated (Table 1). The scenarios varied in duration of disruption, for 3 months (scenarios A) or 6 months (scenarios B), with catch-up of the deficit to either 0% (scenarios 1), 50% (scenarios 2, 4 and 6), or 100% (scenarios 3, 5 and 7). The catch-up targets were met within 3 months (scenarios 2 and 3), 6 months (scenarios 4 and 5) or timed with MMR (approximately 19 months) (scenarios 6 and 7). For the 3-month and 6-month vaccine catch-up scenarios (scenarios 2 to 5), the catch-up was assumed to be linear, whereas for the MMR scenarios (scenarios 6 and 7) the target coverage was reached and timed with the MMR vaccination which is usually given after 26 weeks of age. The worst-case scenario (scenario B1) assumed a 6-month disruption with no catch-up, and the best-case scenario (scenario A3) assumed a 3-month disruption with catch-up to 100% of the vaccine deficit within 3 months.

### 2.6. Model

We used a static mathematical model in Microsoft Excel^®^ (Microsoft Corporation, 2021, Redmond, WA, USA) to estimate the additional number of global TB deaths in the affected birth cohort in the first 15 years of life due to missed and delayed vaccinations, between the baseline (no disruption) and each of the 14 scenarios of disruption. 

The full model has been described in detail elsewhere [7]. In summary, the global number of children at risk of TB death at each week of age t (nt) was calculated as:n_t_ = U_t_ + (V_t_ × (1 − VE)).(4)
where U_t_ is the number of unvaccinated children at each week of age (t), V_t_ is the number of vaccinated children at week t, and VE is the vaccine efficacy against TB death. The estimate for vaccine efficacy against TB death (VE) was based upon published meta-analysis [4]. The global BCG-eligible population was defined as the annual birth cohort from 152 LMIC with a policy of universal neonatal vaccination [16], and the proportion of this global birth cohort receiving BCG by age in weeks (for estimating V_t_ and U_t_) was calculated using data from Clark et al [6]. Background all-cause mortality amongst the birth cohort was accounted for using United Nations, Population Division (UNPD) mortality estimates for children aged 0–15 years disaggregated into 5-year age groups [17]. Estimates for the proportion of children born in a health facility were from the United Nations Children’s Emergency Fund [15]. The estimated reduction in coverage and duration of disruption in each scenario were applied to the proportion of the global BCG-vaccinated birth cohort born within the disruption period.

Risk of TB death was calculated as an age-specific weekly individual risk of tuberculosis death in unprotected children aged 0–4 years and 5–14 years. This risk was based on the number of tuberculosis deaths in the age groups (estimated by WHO [18]) divided by the number of unprotected person-weeks in each age group. Further detail on the calculation for age-specific risk can be found elsewhere [7]. 

This age-specific risk was applied to the number of children at risk (n_t_) in each scenario to estimate the number of TB deaths in the baseline (no disruption to BCG timeliness or BCG coverage based on values from 2019) and in each disrupted scenario. The difference was calculated between each disruption scenario and the baseline scenario. Uncertainty analysis was conducted using Oracle Crystal Ball^®^, with 100,000 parameter sets generated assuming a log normal distribution on the parameters outlined in Table 2. We report the median values for additional number of paediatric TB deaths with 95% uncertainty ranges for each scenario. 

We then compared the average potential benefit of mitigating three different characteristics of COVID disruption: the duration of disruption, the magnitude of catch-up, and the timing of catch-up. The benefit was calculated as the average (mean) number of excess deaths averted in all scenarios with that characteristic, compared to the average (mean) number of excess deaths averted in all scenarios without that characteristic. For example, the benefit of reducing the duration of disruption from 6 to 3 months, was calculated as the mean of the median excess deaths in scenarios B1 to B7 (6 months disruption), compared to the mean of the median excess deaths in scenarios A1 to A7 (3 months disruption).

## 3. Results

### 3.1. BCG Disruption

We obtained disruption data from 29 countries (Table 3). We obtained BCG-specific disruption data from 28 countries, covering 60.0% of the global BCG-eligible population. Data from one country, Indonesia (3.8% of the global BCG-eligible population), indicated disruption to BCG but did not report BCG-specific values, therefore we used estimates of disruption to other routinely delivered paediatric vaccines as a proxy [20]. Of the 28 countries with BCG-specific data, 21 countries reported no disruption to BCG (27.4% of the global BCG-eligible population). Seven countries reported BCG disruptions (32.6% of the global BCG-eligible population), with estimates ranging from 3% to 96% relative reduction in BCG coverage during the disruption period when compared to the same period in 2019 [10,12,21,22]. Of these seven countries, three provided data confidentially.

Using these data, we estimated that the relative global reduction in BCG coverage within the disruption period (R_2020_) was 25% compared to the same period in 2019. Few countries reported the duration of disruption, and where duration was reported the disruption began in March or April 2020, continuing for 3 to 6 months [23]. 

The 14 scenarios (Table 1) were applied to the static mathematical model, to estimate the percentage vaccine coverage of the birth cohort by time since birth, compared to a baseline scenario of no disruption (Figure 1). The scenarios ranged in duration of disruption, the magnitude of catch-up and the timing of the catch-up campaign. Figure 1A and 1B show the modelled timing of BCG in each of the 14 scenarios of disruption. Scenarios B1 to B7, that assumed a 6-month disruption period, resulted in a greater overall reduction and delay in vaccine delivery, compared to scenarios A1 to A7, that assumed a 3-month disruption period.

### 3.2. Impact on Paediatric TB Mortality

The model suggests that any disruption to the current BCG vaccination schedule will lead to an increase in paediatric TB deaths (Figure 2). The worst-case scenario (6-month disruption followed by no catch-up campaigns, B1), may lead to an additional 17.0% (uncertainty range (UR): 0.8–41.6%) or 33,074 (UR: 1506–81,100) paediatric TB deaths, compared to a baseline of no disruption to the BCG vaccine schedule. The best-case scenario (a 3-month disruption period followed by a 100% catch-up within 3 months, A3) may lead to 0.5% (UR: 0–1.1%) or 886 additional TB deaths (UR: 41–2,151). 

This modelling exercise assessed three main characteristics related to COVID-19 related disruption: duration of disruption, magnitude of catch-up, and timing of catch-up. The magnitude of catch-up was found to be the most influential variable in minimising excess paediatric TB mortality. Complete catch-up of BCG amongst children who have missed their vaccines due to COVID disruption (scenarios A3, A5, A7, B3, B5, and B7), was estimated to avert an average of 21,930 excess deaths, when compared to scenarios without catch-up (scenarios A1 and B1). 

The duration of disruption was found to be the second most influential characteristic assessed in our model. A three-month disruption period was found to avert an average of 7295 excess deaths (scenarios B1 to B7), when compared to scenarios with a six-month disruption period (scenarios A1 to A7).

Reducing the length of time taken to meet the catch-up target was found to be the least influential characteristic. Taking 3 months to catch-up was found to avert an average of 1612 excess deaths (scenarios A2, A3, B2 and B3), when compared to scenarios with catch-up over 19 months (as defined by the MMR schedule) (scenarios A6, A7, B6 and B7).

## 4. Discussion

We estimated a global average of 25% reduction in BCG coverage within the COVID-19-related disruption period, with countries reporting a reduced coverage of between 0 and 100% compared to BCG coverage in the same period in 2019. We estimated this could result in an additional 886 (UR: 41–2151) paediatric TB deaths, for a 3-month disruption period followed by catch-up to previous BCG coverage within 3 months, to an additional 33,074 (UR: 1506–81,100) paediatric TB deaths if disruption continued for 6 months with no catch-up.

We explored a variety of scenarios for the duration of disruption, and the timing and coverage of catch-up. The results indicate that maximising the coverage of BCG catch-up is especially important for averting paediatric TB deaths. Vaccinating all children who have missed their BCG vaccines could prevent an average of 21,930 paediatric TB deaths. Little is currently known about the duration of COVID-19-related disruptions as few countries have reported on this. Our analysis suggests that minimising the duration of disruption by even 3 months can avert an average of 7295 additional paediatric TB deaths. Rapid action is required to mitigate and minimise the burden of paediatric TB mortality that will otherwise inevitably take place in many settings. 

Ensuring that BCG vaccination coverage is recovered following periods of disruption is an important intervention and will minimise the excess paediatric TB mortality in the coming years. However, catch-up campaigns can be costly and logistically challenging. We used the timing of MMR vaccination as an example of providing BCG catch-up alongside other routine vaccines, that could minimise cost and logistics of catching up BCG coverage. Our analysis showed that the coverage of catch-up may be more important than the timing of the catch-up, where reducing the catch-up period from 19 months (MMR timing) to 3 months only averts 1612 excess deaths. Therefore, children could receive the BCG vaccine when they attend health facilities for other routine check-ups or vaccines. In many BCG-scarce settings a vial of BCG is only opened if enough children are present for vaccination, thus creating missed opportunities for vaccination when few children are present [14,24]. Providing sufficient BCG vials to clinics could be an effective way of ensuring timely BCG vaccination and catch-up, and thereby minimising excess paediatric TB deaths. 

Given that COVID-related disruptions may also impact the ability to catch up BCG vaccination through routine visits, other options may also need to be considered to help minimise the length of disruptions and facilitate catch-up vaccination. Examples include increasing access by providing vaccines in clinics, separated in time or location from unwell patients, drive-through clinics, or door-to-door vaccination provision, and by increasing demand through public health messaging on the importance of continuing timely routine vaccination during lockdowns [10]. Political motivation and global support are required to support health programs in overcoming logistical challenges related to catch-up strategies for BCG. 

### 4.1. BCG in the Fight against Paediatric TB

COVID-19 has substantially disrupted general health services, especially in countries where TB is endemic [25]. WHO estimates 1.4 million fewer people received TB care in 2020 and a 28% shortfall in TB cases diagnosed when compared with 2019 [26]. Previous modelling studies estimated that reduced TB healthcare seeking behaviour and access to appropriate TB diagnostics and treatment caused by the COVID-19 pandemic could lead to a net increase in TB cases and deaths in the coming years when social distancing is eased [27]. 12% of TB cases in 2019 were amongst children below 15 years of age [18]. As BCG remains the only approved vaccine against TB, maintaining BCG vaccination coverage is important to protect paediatric populations against a potentially increased risk of TB in the post-COVID-19 era. Aside from the BCG vaccine’s well-documented effects on miliary and meningeal TB, increasing evidence suggests that BCG may offer protection against other non-related diseases and all-cause mortality in children [28,29], increasing the importance of minimising BCG disruptions. 

### 4.2. Why Has BCG Been Impacted?

A recent survey in South-East Asia and Western Pacific noted fear of infection, restriction of movement, and limited access to healthcare as key reasons for COVID-related disruption to immunisation services in 2020 [10]. As hospitals struggle to manage COVID-19 patients with limited resources, the proportion of births occurring outside of healthcare settings are also likely to have increased, which may contribute to delays in routine immunisations that are given at birth, including BCG [30]. As the pandemic progresses, COVID-19 transmission and fear of infection remains very high in certain regions, potentially disrupting immunisation services for a longer period than we have estimated in this study [31]. In India, one of the largest producers and consumers of BCG vaccines, COVID-19 cases and mortality climbed at an unprecedented rate in mid-2021 [15,32]. While in this study we consider a continuous disruption to BCG immunisations for a limited period, a changing reality suggests recurring disruptions and intermittent localised lockdowns will continue well into 2021 and onwards for many countries.

Gavi identifies 68 countries where immunisation services have been affected due to the COVID-19 pandemic [1]. Although we use MMR as a ‘realistic’ case scenario for delivering catch-up BCG vaccination, MMR and other routine immunisations have also been severely impacted due to COVID-19, therefore efforts will be needed to ensure coverage rates recover across all immunisations [10,33]. UNICEF noted a 70–80% reduction in vaccine shipments due to air freight restrictions, and many aviation services in Africa are bankrupt or not running due to the pandemic [3,34]. Countries which struggle to pay the additional airfare costs to regain shipments are likely to experience stockouts. Aside from reduced demand for BCG during the COVID-19 pandemic, there are severe threats to maintaining the level of supply required to sustain adequate vaccination schedules for neonates, disproportionately affecting low-income countries. Global action is required to ensure swift supply of vaccines to countries most disrupted by the COVID-19 pandemic.

### 4.3. Other Threats to BCG Coverage

BCG vaccination supply is already highly sensitive to stockouts, and fluctuations caused by instability in the production market [35]. A large shortfall of BCG production due to technical difficulties in 2015 led to a gap of 16.5 million vaccines. Modelling estimated 7433 (95 % UR: 320–19,477) additional paediatric TB deaths because of this interruption to vaccine supply [9]. Importantly, 24,914 (UR: 1074–65,278) additional deaths may have been avoided due to prompt shortfall reduction measures. The 2019 UNICEF SD Market Dashboard notes BCG supply as moderate to fragile, due to the delisting of a major producer which represented a 30% reduction of global BCG vaccine supply availability [36]. There is now a reliance on four pre-qualified BCG manufacturers. One of these, the Serum Institute of India, is reported to be under immense pressure to meet COVID-19 vaccine production targets, caused by restraints in raw materials and resources [37,38]. COVID-19 vaccine production may be more profitable than other vaccines and political incentives for reaching high production of COVID-19 vaccines may be exacerbating a switch in vaccine production priorities.

Furthermore, several studies have explored the potential for BCG to have non-specific beneficial effects against COVID-19 disease [39]. Recently published results by Tsilika et al., found that BCG revaccination resulted in 68% risk reduction for COVID-19 disease [40]. Although these results are scientifically interesting, they are based on very few cases. If such findings were to lead to redirection of BCG from newborns to healthcare workers or older people in settings where COVID-19 vaccination rates are low, BCG availability for neonatal vaccination could be impacted. 

### 4.4. Limitations

One of the key limitations of this study is the lack of good estimates for BCG efficacy against TB death, we used estimates from a meta-analysis on the effectiveness of BCG with wide confidence intervals (8–88%) [4]. Research on the effectiveness of BCG against TB death is limited and must be explored further. Additionally, this study was limited in scope to a high-level global estimate, and therefore a global estimate for the risk of TB death amongst under-15s was used. We used TB mortality estimates for children aged 0–15 years and assumed no waning in BCG efficacy over time, thereby potentially underestimating the additional number of paediatric TB deaths. 

We chose to use a static model and therefore our model does not account for any potential reduction in transmission due to decrease in disease burden. However, any potential effects are likely to be small as paediatric tuberculosis cases are minimally infectious.

Hospitals have struggled to maintain health services during the pandemic, therefore our estimate of births taking place in health facilities may be an overestimate. In addition, stock-outs may affect the number of vaccinations taking place in health facilities at the time of birth. Our results therefore likely underestimate the level of impact on BCG due to COVID-19 related interruptions. 

Data were only available for 29 countries covering 64% of the BCG-eligible population, but our estimate for BCG disruption carries some uncertainty due to missing data. We found substantial variation in the magnitude and duration of COVID-19-related disruption by country and by month, with disruption continuing periodically through 2020 and into 2021. Therefore, our estimates of the duration of disruption are likely to be an underestimate. It will be necessary to re-evaluate these results when updated and consistent global data on the number of BCG vaccines delivered in 2020 and 2021 are available. 

In this model, we did not account for the non-specific effects of BCG, and the potential impact of BCG disruptions on excess paediatric mortality may be even larger. A country-specific model estimating the importance of delays in BCG vaccination found that delays in BCG were particularly important in the first period of life, where all-cause mortality is highest [14]. Thus, the impact of timing of catch up could potentially be greater than was found in this study if the impact of BCG on all-cause mortality were included and more granular TB mortality data were available.

## 5. Conclusions

Ensuring catch-up vaccination of children who have missed their scheduled vaccines is a critical immunisation priority to minimise excess paediatric TB deaths. Our results suggest that high levels of catch-up vaccination coverage, in addition to minimising the duration of disruption, should be a priority to minimise additional paediatric TB deaths due to COVID-related disruptions. BCG catch-up alongside other routine vaccines or at other health facility contacts may be a feasible way to deliver vaccines without separate campaigns. Global action is required to support the most disadvantaged countries in overcoming the logistical challenges this intervention will entail, and action is required to support countries with recovering vaccination coverage as soon as possible.

## Figures and Tables

**Figure 1 vaccines-09-01228-f001:**
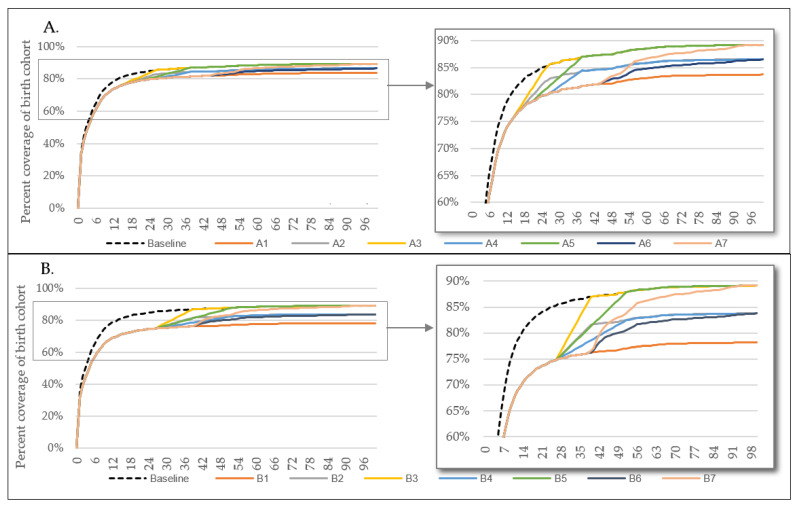
Vaccine coverage scenarios by age in weeks compared to baseline of no disruption; (**A**) Scenarios A1 to A7 e.g., disruption for 3 months; (**B**) Scenarios B1 to B7 e.g., disruption for 6 months.

**Figure 2 vaccines-09-01228-f002:**
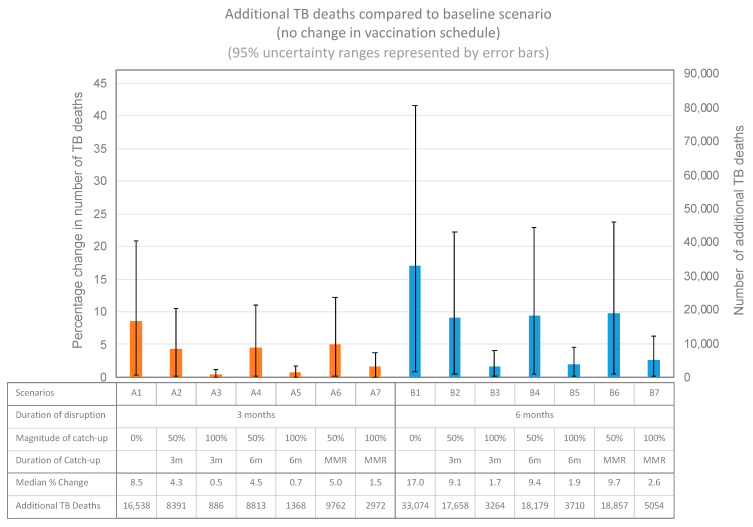
Additional paediatric tuberculosis (TB) deaths from 14 scenarios compared to a baseline of no disruption.

**Table 1 vaccines-09-01228-t001:** Summary of the disruption scenarios ranging in duration, timing, and magnitude of the catch-up.

Scenario	A	B
Disruption period	Disruption period: 3 m (e.g., April 2020 to June 2020)	Disruption period: 6 m (e.g., April 2020 to September 2020)
Catch-up strategy	1. None 2. Catch-up: 50% of deficit–in 3 months 3. Catch-up: 100% of deficit–in 3 months 4. Catch-up: 50% of deficit–in 6 months 5. Catch-up: 100% of deficit–in 6 months 6. Catch-up: 50% of deficit–timed with MMR ^1^ 7. Catch-up: 100% of deficit–timed with MMR ^1^	1. None 2. Catch-up: 50% of deficit–in 3 months 3. Catch-up: 100% of deficit–in 3 months 4. Catch-up: 50% of deficit–in 6 months 5. Catch-up: 100% of deficit–in 6 months 6. Catch-up: 50% of deficit–timed with MMR ^1^ 7. Catch-up: 100% of deficit–timed with MMR ^1^

^1^ Measles, mumps, and rubella (MMR).

**Table 2 vaccines-09-01228-t002:** Data inputs used for developing uncertainty analysis.

Parameters for Uncertainty Analysis	Point Estimate	Uncertainty Interval	Reference
Vaccine efficacy against tuberculosis death	0.66	0.08–0.88	[4]
HIV-negative paediatric male tuberculosis deaths in 2019	104,000	93,000–115,000	[18]
HIV-negative paediatric female tuberculosis deaths in 2019	90,000	80,000–99,000	[18]
Tuberculosis deaths in children younger than 5 years who had not received tuberculosis treatment in 2015 (HIV negative)	161,000	108,000–223,000	[19]
Tuberculosis deaths in children younger than 5 years who had received tuberculosis treatment in 2015 (HIV negative)	2690	1850–4180	[19]
Tuberculosis deaths in children aged 5–15 years who had not received tuberculosis treatment in 2015 (HIV negative)	31,500	18,600–51,400	[19]
Tuberculosis deaths in children aged 5–15 years who had received tuberculosis treatment in 2015 (HIV negative)	2050	1510–3100	[19]

**Table 3 vaccines-09-01228-t003:** Summary of data gathered on global BCG disruption in 2020 due to COVID-19 (*n* = 29 countries).

Country	Percentage of Global BCG ^1^ Vaccine Coverage Delivered by Each Country (2019)	Reduction in BCG ^1^ Coverage during Disruption (2020 vs. 2019)	Reduction in Coverage of Routine Paediatric Immunisations during Disruption (2020 vs. 2019)	References
India	19.6%	50%		[21]
China	14.5%	0%		[10]
Pakistan	4.6%	41%		[12]
Indonesia	3.8%		26%	[20]
Bangladesh	2.5%	96%		Country source
Democratic Republic of Congo	2.2%	0%		[13]
Brazil	2.0%	0%		Country source
Philippines	1.4%	0%		[10]
Other countries (*n* = 21) ^2^	13.2%	8% ^3^		
WEIGHTED AVERAGE:		25%		

^1^ Bacillus Calmette-Guérin (BCG); ^2^ Data from confidential sources and smaller countries where percentage of global vaccine coverage was below 1%; ^3^ Average across these countries.

## Data Availability

Not applicable.

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
