# Peer review of "Impact of COVID-19 Disruptions on Global BCG Coverage and Paediatric TB Mortality: A Modelling Study"

_vaccines, 2021, doi:10.3390/vaccines9111228_

Round 1

Reviewer 1 Report

The present MS could NOT be typical academic, especially so called pure science.  However, we have to push this to publish and to open for people ASAP, without any hesitation.     

Author Response

Thank you for your feedback.

Reviewer 2 Report

This is a very well written manuscript that deals with the consequences of the current Covid19 pandemic on BCG vaccination worldwide. There are surely some weaknesses on this kind of study – they are predictions – and there is not a direct correlation between the lack of BCG vaccination and TB mortality but these points are stressed by the authors. In summary, a nice article focusing on an important topic and submitted to the correct journal.

Author Response

Thank you for your kind feedback.

Reviewer 3 Report

In this paper, the authors estimated the impact of COVID-19 disruptions on global BCG coverage and on paediatric TB mortality in low and middle income countries with a universal BCG policy, in order to help guide public health decision making around mitigation measures.

The paper can be accepted for the publication after some minor revisions.

  1. All acronym used in this paper should be defined.
  2. Give more motivation and novelty to your study.
  3. Add the punctuation to each mathematical equations.
  4. There are some typos. The authors should carefully read the manuscript.

Author Response

Thank you for your feedback, the suggestions have been taken on board.

  1. We have defined the acronyms.
  2. We have also added further clarification to the introduction to explain why the study is novel and necessary, please see lines 115 to 124 of the manuscript for the amendments.
  3. We have added punctuation to the mathematical equations.
  4. We have reviewed the manuscript.

Reviewer 4 Report

The author estimated the impact of COVID-19 disruptions on worldwide BCG coverage and paediatric tuberculosis mortality in LMICs with a universal BCG policy in order to inform public health decision-making regarding mitigation strategies. BCG catch-up in conjunction with other standard vaccines may be a viable strategy for achieving catch-up without conducting separate campaigns. The manuscript is well structured and well discussed. However, some points should be checked and corrected before its acceptance in this journal.  Therefore, I recommended the publications of the paper after minor revision according to given my comments.

  • Please provide the expansion of abbreviations in abstract - Bacillus Calmette-Guérin (BCG). Many abbreviations should be expanded.
  • In material and methods, please provide the statistical analysis.
  • In figure, please provide the statistical significance.
  • In Conclusion, the authors should add the significance of this research, and its potential practical application.

Author Response

Thank you for your comments and feedback.

  1. We have expanded all abbreviations
  2. We have added further clarification to the methods to explain the statistical analysis conducted in this study, please see lines 257-272.
  3. We do not conduct statistical tests to compare the scenarios. The uncertainty ranges represent the credible interval for the range in excess deaths in each scenario. The likely difference between scenarios is represented by the difference in the point estimates. The uncertainty ranges are very wide due to the uncertainty in the vaccine effectiveness.
  4. We have already stated the following practical advice in the conclusion, "Our results suggest that high levels of catch-up vaccination coverage, in addition to minimising the duration of disruption, should be a priority to minimise additional paediatric TB deaths due to COVID-related disruptions. BCG catch-up alongside other routine vaccines or at other health facility contacts may be a feasible way to deliver vaccines without separate campaigns." Given the analysis in this study is quite high-level, we believe this level of detail in this practical recommendation is sufficient.